# Topical Neuropeptide Y for Ischemic Skin Wounds

**DOI:** 10.3390/ijms25063346

**Published:** 2024-03-15

**Authors:** Tais Stangerup, Lise Mette Rahbek Gjerdrum, Michael Bzorek, Line Andersen, Anne-Marie Heegaard, Lars N. Jorgensen, Magnus S. Ågren

**Affiliations:** 1Digestive Disease Center, Bispebjerg Hospital, University of Copenhagen, 2400 Copenhagen, Denmark; tais.dk@gmail.com (T.S.); lars.nannestad.joergensen@regionh.dk (L.N.J.); 2Department of Pathology, Zealand University Hospital, 4000 Roskilde, Denmark; lmgj@regionsjaelland.dk (L.M.R.G.); mibz@regionsjaelland.dk (M.B.); 3Department of Clinical Medicine, Faculty of Health and Medical Sciences, University of Copenhagen, 2200 Copenhagen, Denmark; 4Department of Drug Design and Pharmacology, Faculty of Health and Medical Sciences, University of Copenhagen, 2100 Copenhagen, Denmark; line.andersen.email@gmail.com (L.A.); amhe@sund.ku.dk (A.-M.H.); 5Department of Dermatology, Copenhagen Wound Healing Center, Bispebjerg Hospital, University of Copenhagen, 2400 Copenhagen, Denmark

**Keywords:** growth factors, neuropeptides, release diffusion model

## Abstract

Our objective was to investigate the effects of topically applied neuropeptide Y (NPY) on ischemic wounds. Initially, the animal model for ischemic wound healing was validated using 16 male Sprague Dawley albino rats. In the intervention study, an additional 28 rats were divided into three groups: NPY (0.025%), the positive control insulin-like growth factor-I (IGF-I, 0.0025%), and the hydrogel carrier alone (control). The hydrogel was selected due to its capacity to prolong NPY release (*p* < 0.001), as demonstrated in a Franz diffusion cell. In the animals, an 8 mm full-thickness wound was made in a pedunculated dorsal ischemic skin flap. Wounds were then treated and assessed for 14 days and collected at the end of the experiment for in situ hybridization analysis (RNAscope^®^) targeting NPY receptor *Y2R* and for meticulous histologic examination. Wound healing rates, specifically the percentage changes in wound area, did not show an increase with NPY (*p* = 0.907), but there was an increase with rhIGF-I (*p* = 0.039) compared to the control. *Y2R* mRNA was not detected in the wounds or adjacent skin but was identified in the rat brain (used as a positive control). Light microscopic examination revealed trends of increased angiogenesis and enhanced inflammatory cell infiltration with NPY compared to control. An interesting secondary discovery was the presence of melanophages in the wounds. Our findings suggest the potential of NPY to enhance neovascularization under ischemic wound healing conditions, but further optimization of the carrier and dosage is necessary. The mechanism remains elusive but likely involves NPY receptor subtypes other than Y2R.

## 1. Introduction

Wound management poses a considerable challenge for patients and contributes significantly to healthcare costs [1,2]. Ischemia, often associated with conditions like diabetes mellitus, hampers the natural process of wound healing. While various treatments exist, none have demonstrated convincing clinical efficacy under ischemic conditions [3].

Peptide growth factors play a crucial role in wound healing [4], and the significance of neuropeptides in this context has been acknowledged [5,6]. Neuropeptide Y (NPY), a 36-amino acid peptide, is the most abundant neuropeptide in the brain, participating in both central and peripheral nervous processes. Centrally, NPY regulates appetite, feeding, stress response, anxiety, circadian rhythms, and blood pressure [7,8]. It is associated with the adrenergic mediators along the terminal ends of sympathetic nerve fibers. NPY stimulates endothelial cell proliferation and migration and angiogenesis [9,10,11,12], which is particularly relevant for ischemic wound healing.

Humans have four different G-protein coupling functional NPY receptor subtypes (Y1R, Y2R, Y4R, and Y5R) [8,13]. Y4R is predominantly expressed in the colon, small intestine, and prostate, preferentially activated by pancreatic polypeptide (PP). Conversely, Y1R, Y2R, and Y5R exhibit high affinity for NPY [8]. Y1R is highly expressed in smooth muscle cells and mediates vasoconstriction [14], including in the skin [15]. The presence and function of Y2R and Y5R in the skin are not well understood [16]. In the cornea of mice, the expression of Y1R, Y2R, and Y5R was differentially studied in induced angiogenesis through immunohistochemical staining [10]. Y2R, but not Y1R or Y5R, was detected on the formed blood capillaries [10]. In full-thickness skin wounds made on the back of Y2R-deficient mice, topical NPY neither stimulated angiogenesis nor restored delayed wound healing, unlike in wild-type mice [10]. These results underscore the importance of Y2R in mediating the beneficial effects of NPY on angiogenesis and wound healing in normoxia.

The situation appears different under ischemic conditions, where Y2R protein levels were increased in ischemic compared to nonischemic full-thickness skin wounds in rabbits [17]. In rats, *Y2R* mRNA was absent in nonischemic skeletal muscle but induced in ischemic muscle, and exogenous NPY promoted angiogenesis specifically in ischemic muscle [18,19]. Notably, there are no studies investigating the effect of NPY on cutaneous wound healing under ischemic conditions in rats or other animal species, and the expression of NPY receptors in skin and skin wounds needs further clarification [10].

The delivery system for growth factors is crucial [20,21,22]. Hydrogels, composed of three-dimensional polymeric materials, swell upon contact with water, and their mechanical properties determine the release profile of pharmaceutical agents [23]. Unfortunately, there is no standardized laboratory test to evaluate the impact of hydrogels on the release of bioactive molecules [24,25].

In this study, we explored the release of NPY in an amorphous hydrogel in vitro using a Franz diffusion cell [25]. Furthermore, we investigated the effects of NPY in vivo on ischemic wound healing in a validated preclinical model [21,26]. Insulin-like growth factor-I (IGF-I), known to promote wound healing in this model, was used as a positive control [21]. Wound areas were measured until termination postoperative day 14, when biopsies were taken for in situ hybridization analysis of *Y2R* mRNA using the RNAscope^®^ technology and for histological examination.

## 2. Results

The primary objective of these studies was to examine the impact of NPY on ischemic wounds. Initially, we validated the ischemic wound model. Subsequently, we assessed the release of NPY from a hydrogel designed for sustained delivery using a Franz Cell in vitro. Finally, we utilized this hydrogel carrier to investigate the effect of locally delivered NPY to ischemic wounds in vivo.

### 2.1. Validation of the Ischemic Wound Model

The ischemic wound model’s validation involved comparing wound area changes in the skin flap (ischemic wound) with those in adjacent normal skin (nonischemic wound) outside the skin flap, made using an 8 mm punch biopsy instrument. The relative change in wound area from day 0 to day 6 revealed a slower healing process for the ischemic wounds, irrespective of the topical treatment, in comparison to the nonischemic wounds within the same animal (Figure 1). The hydrogel carrier alone did not influence the healing of either the ischemic (*p* = 0.603) or the nonischemic wounds (*p* = 0.748).

### 2.2. Release of NPY In Vitro

When NPY was introduced into PBS, the donor chamber exhibited a gradual depletion of NPY, with less than 50 µL of donor solution remaining after 6 h. The amorphous hydrogel containing NPY initially displayed slight acidity but became neutral after about 2 h (Figure 2a). At the 6 h mark, an amount of 250 ± 8.1 (mean ± SD) mg material was aspirated from the donor chamber. The rheology of the hydrogel remained essentially unaltered. The release of NPY from the hydrogel (0.4–0.5 µg per hour) over the 6 h experimental period was significantly (*p* < 0.001) reduced compared to NPY in the PBS (Figure 2b).

### 2.3. Effects of NPY In Vivo

#### 2.3.1. Body Weight Development

A significant (*p* < 0.001) effect of time on postoperative body weight changes was observed. The animals experienced a weight loss of approximately 4% from day 0 to day 2 (*p* = 0.007) but showed weight gain of around 6% from postoperative day 8 (*p* < 0.001) compared with their preoperative body weight on day 0. No significant (*p* = 0.572) effect of the treatments on body weight development was found.

#### 2.3.2. Ischemic Wound Healing

Wound healing, assessed as the relative change in wound area from day 0, followed the expected pattern based on previous results from the applied ischemic wound model [21,26]. The wound healing curves indicated significant effects of postoperative time (*p* < 0.001) and treatment (*p* = 0.025). Overall, the wounds decreased in size from postoperative day 6 compared to their initial size on the day of wounding (*p* = 0.032). There was no significant difference in wound healing (*p* = 0.907) between the NPY and the hydrogel control group, while the positive control topical IGF-I increased wound healing compared to the hydrogel control (*p* = 0.039), as illustrated in Figure 3.

#### 2.3.3. In Situ Hybridization for Y2R mRNA

*Y2R* mRNA could not be detected in the wounds or in adjacent uninjured skin. In another animal experiment involving three-day-old 6 mm biopsy nonischemic wounds on the back of rats [27], *Y2R* mRNA was also not detected, while scattered ganglion cells in rat colon were positive for *Y2R* mRNA by the RNAscope^®^ assay.

#### 2.3.4. Light Microscopy Examination

Representative microphotographs of the wounds stained with hematoxylin and eosin (H&E) are shown at two magnifications in Figure 4. Angiogenesis was most pronounced in the NPY-treated wounds. Lymphocytes, constituting approximately 75% of immune cells in the granulation tissue, were the predominant cell type, followed by neutrophil granulocytes (about 20%). The distribution among the different immune cells was similar in the three groups, but they were more numerous in the NPY group. Histiocytes were not quantified, as they could not definitively be distinguished from fibroblasts in the H&E-stained sections [28]. Mast cells were few in numbers. Dermal melanophages were identified in all groups but were not quantified. Quantifying collagen in van Gieson-stained sections proved challenging, but the control wounds appeared to contain the most collagen among the three groups. The ingrowth of epithelium in the NPY-treated and IGF-I-treated wounds appeared to be more advanced than in the control-treated wounds (Table 1). Foreign bodies were observed in all wounds, accompanied by multinucleated giant cells in the NPY and IGF-I groups. The appearance of a multinucleated giant cell is shown in Figure 4f.

## 3. Discussion

This study aimed to investigate the impact of repeated topical application of the neuropeptide NPY on ischemic full-thickness skin wounds in a standardized animal model. Despite NPY treatment showing an increase in new blood vessel formation, analogous to the observed stimulatory effects on cultured endothelial cells [11], this heightened angiogenesis did not correlate with accelerated wound healing rates.

NPY activates cells through four main functional receptors. We opted to examine the presence of Y2R because findings from multiple studies have suggested that Y2R plays a crucial role in mediating NPY-induced angiogenesis and is up-regulated under ischemic conditions [10,12,18,29]. Unexpectedly, using the RNAscope^®^ technique, *Y2R* mRNA could not be detected in the wounds, but it was detected in neurons as expected [30]. While Pradhan et al. [17] detected *Y2R* mRNA in rabbit full-thickness skin wounds using qRT-PCR, the mRNA levels in the ischemic wounds were not increased compared to those in nonischemic wounds. It is possible that our assay was less sensitive than theirs. Silva et al. [31] treated endothelial cells (HUVEC) with proinflammatory cytokines and observed a significant downregulation of *Y2R* mRNA, while *Y1R* mRNA remained unchanged. This observation is relevant, considering that proinflammatory cytokine levels are elevated in ischemic wounds [32]. The absence of *Y2R* expression may also be attributed to the gradual vascularization of the skin flap from underneath, transitioning it from ischemic to nonischemic tissue. Notably, intracutaneous pO_2_ levels did not differ between the skin flap and adjacent normal skin on postoperative day 14 in the animal model employed in this study [21].

In contrast to NPY, topical application of IGF-I stimulated the healing of the ischemic wounds over 14 days, consistent with previous results in this animal model [21]. Beckert et al. [21] conducted a study applying IGF-I daily using two different carriers and found that IGF-I in a PVA film was more effective than IGF-I delivered in a methylcellulose gel, supporting the notion that the mode of peptide delivery is crucial for their stimulatory effect. Our goal was to achieve sustained delivery of peptides, assuming that this would prolong their bioactivities in the wounds. The selected hydrogel has been used alone or as a carrier for bioactive compounds in rodent wound healing studies [33,34,35,36,37,38] and met our objective by delaying the release of the NPY depot, at least in the in vitro Franz Cell setup. The mechanisms contributing to sustained release are likely multiple, such as electrostatic interaction between the basic NPY with positive net charge at pH less than 7.0 [39] and the polyanionic polymer carboxymethyl cellulose (CMC) of the hydrogel, along with diffusion resistance.

Given the exploratory nature of this study, a relatively high dosage of NPY was chosen. Ekstrand et al. [10] delivered 1.5 µg NPY to 6 mm biopsy wounds using pellets made of sucralfate and poly(2-hydroxyethyl methacrylate). Sucralfate inhibits pepsin activity [40], potentially increasing the half-life of NPY in wounds. Their dosage should be compared with ours, which involved delivering 50 µg NPY to 8 mm wounds. The in vitro release experiment indicated a delivery rate of 0.4–0.5 µg NPY per hour. Additionally, the molecular forms that the delivered full-length NPY_1–36_ adopts in the proteolytic microenvironment of the wounds remain unknown. In human serum (25–50 ng NPY/mL), proteinases generate NPY species with different receptor affinities [41]. The dominant species NPY_3–36_ is generated by dipeptidyl peptidase IV (DPPIV), implicated in angiogenic processes during wound healing [42]. In summary, there is a need for optimization of topical NPY administration to wounds.

One drawback observed was the unfavorable tissue reactions to the hydrogel, evidenced by the presence of foreign bodies and multinucleated giant cells in the granulation tissue. CMC is a common constituent in hydrogel, film, and fiber wound dressings [43]. For instance, the growth factor rhPDGF-BB is formulated in aqueous sodium CMC as Regranex^®^ 0.01% gel, and no foreign bodies were observed in rat wounds treated with nonmodified sodium CMC [44]. In contrast, wound treatment with another brand of amorphous hydrogel composed of modified (crosslinked) CMC resulted in foreign bodies with associated multinucleated giant cells [45], highlighting the impact of chemical modification on the biocompatibility and biodegradability of CMC. We do not have information on the structure of the CMC in the hydrogel used here. It is recognized that the viscosity of hydrogels increases with the degree of crosslinking. The viscosity of the hydrogel matches that of a crosslinked CMC [46,47]. This brand is approved by FDA for the use in tissue expansion [46]. Although the hydrogel was slightly diluted (5%) and acidified (pH~6.4), this modification is unlikely to have caused the adverse tissue reactions, as Pilakasiri et al. [48] reported similar tissue reactions to the hydrogel in full-thickness excisions in normoxic skin. They observed adverse consequences of the hydrogel residues, particularly on epithelialization during the later stages of wound healing, as foreign bodies in the granulation tissue may impair keratinocyte migration [49]. This may explain our findings of no effect of the hydrogel on early wound healing up to postoperative day 6, i.e., wound contraction, but less epithelialization than expected based on previous observations in the same animal model [26]. Taken together, hydrogel residues may have impeded epithelialization and potentially masked beneficial effects on epithelialization during the later stage of wound healing.

NPY treatment led to an increase in the number of immune cells in the wounds. We speculate that the heightened infiltration of inflammatory cells is not the consequence of increased deposition of foreign bodies. The immunomodulatory effects of NPY are complex, and both anti-inflammatory and pro-inflammatory properties have been attributed to NPY [13,50].

A notable discovery was the presence of melanophages in the wounds. The literature provides inconsistent information on whether melanin is present in the skin of albino rats [51,52,53]. Our histology results indicate that wounds of albino rats do contain melanin, although the sources of melanin are currently unknown. One possibility is that microorganisms colonizing the wounds may have produced melanins [54].

Collectively, our findings imply that NPY has the potential to improve ischemic wound healing, but further work is required to optimize the dosing regimen.

## 4. Materials and Methods

### 4.1. Chemicals and Reagents

Synthetic NPY (rat, human) with a peptide purity exceeding 96% by high-performance liquid chromatography and a molecular weight of 4271 Da was used for the in vitro and in vivo studies (H-6375, Bachem AG, Bubendorf, Switzerland) as a trifluoroacetate salt. rhIGF-I (molecular weight: 7600 Da) was purchased (Cat. No.: 100-11) from PeproTech (Rocky Hill, NJ, USA). The amorphous, hypertonic [47] hydrogel (IntraSite^®^ Gel, Smith & Nephew, Hull, UK) consists of 2.3% modified CMC, propylene glycol (20%), and water. The pH of the hydrogel was 6.8 measured by a gel-filled pH/ATC (automatic temperature compensation) double-junction combination electrode (13-620-111) connected to an accumet™ AB315 pH meter (Fisher Scientific Company, Pittsburgh, PA, USA).

### 4.2. In Vitro Release of NPY

The release of NPY from the hydrogel was studied using a flow-type Franz Cell in borosilicate glass (PermeGear, Hellertown, PA, USA). The Franz Cell with flat ground joint comprises a donor chamber (9 mm inner orifice diameter, 1.0 mL) and a receptor chamber (5.5–5.7 mL). A porous (0.22 µm), 165–185 µm thick Millipore Express^®^ (Merck KGaA, Darmstadt, Germany) hydrophilic polyethersulfone membrane separated the donor and receptor chambers. To prevent leakage, the apical side of the membrane was sealed to the donor chamber using transparent silicone adhesive (Dowsil™ 732, Dow Europe GmbH, Wiesbaden, Germany), and a Teflon foam ring was placed in the interface between the donor and receptor chambers. The two chambers were held together by a clamp (Figure 5).

The donor and receptor chambers were filled with Dulbecco’s phosphate-buffered saline, pH 7.4 (PBS), and the diffusion cell was equilibrated for 2 h at 37.0 °C. The temperature was maintained at 37.0 °C using circulating water in the heating jacket from an external water bath (Haake S5P, Thermo Fisher Scientific, Newington, NH, USA). The receptor chamber was continuously agitated with a magnetic Teflon-coated cylindrical (8 mm × 3 mm) stirring bar. A total of 10 µL NPY (5 mg/mL 10 mM acetic acid (HOAc)) was mixed with 190 µL of the hydrogel and centrifuged at 2000× *g* for deaeration. The development of pH of the blend was studied in a separate experiment using the phenol red pH indicator (final concentration: 0.015 mg/mL; P3532, Sigma-Aldrich, St. Louis, MO, USA). The control consisted of 10 µL NPY (5 mg/mL 10 mM HOAc) and 190 µL PBS. The 200 µL blends were added to the donor chambers using a positive displacement pipette (Microman, Gilson, Middleton, WI, USA). The donor compartment was covered with Parafilm^®^ M (American National Can™, Chicago, IL, USA). Aliquots (200 µL) of receptor medium were obtained after 0.5, 1, 2, 4, and 6 h using a PermeGear pipette tip (SES GmbH, Bechenheim, Germany) via the upper sampling arm and replaced with fresh PBS via the lower sampling arm. At the end of the experiment, the contents of the donor chamber were emptied with the positive displacement pipette and weighed. The samples were added to 200 µL of the reagent diluent (1% BSA in PBS) used in the ELISA assay [25] and stored in 1.5-mL polypropylene reaction tubes (Cat. No.: 616201, Greiner Bio-One, Frickenhausen, Germany) at −80 °C until analyzed on NPY contents by a DuoSet™ ELISA (DY8517-05, R&D Systems, Minneapolis, MN, USA) with ancillary reagents (DY008B, R&D Systems). The cumulative amount of NPY (µg) released was calculated for each cell from a standard curve fitted by 4-parameter logistic function (SigmaPlot 14.0, Systat Software, Inc., San Jose, CA, USA).

### 4.3. In Vivo Wound Experiments

The experiments were conducted in adherence to institutional guidelines for care and use of laboratory animals, with approval from the Danish animal ethical committee (2006/561-1092).

#### 4.3.1. Animals

Forty-four male 9-week-old Sprague Dawley albino rats (Taconic M&B, Borup, Denmark), weighing 280–330 g, were utilized. The animals were provided a basal diet (Altromin 1324, Altromin Spezialfutter, Lage, Germany) ad libitum and individually housed in standard type IV cages at controlled temperatures (18–22 °C) and light (12 h light/12 h dark, lights on from 06.00 a.m.) [55]. A week of acclimatization was allowed before surgery.

#### 4.3.2. Surgery and Wounding Day 0

Preoperative analgesia (0.1 mg/kg buprenorphine) was administered subcutaneously, and rats were anesthetized via subcutaneous injection of a mixture containing 0.14 mg/kg fentanyl citrate, 4.4 mg/kg fluanisone, and 2.2 mg/kg midazolam [55].

The rats’ backs were shaved, cleansed with 70% alcohol, and a standardized pedunculated dorsal ischemic skin flap (3 cm × 7 cm) was lifted, repositioned, and fixed with skin staples [21]. An 8 mm trephine was used to make one full-thickness wound in the ischemic skin flap, and in the validation study, another full-thickness wound was made in normal skin outside the ischemic skin flap (Figure 1). The outer margins of the wounds were traced on a transparent sheet (Opsite Flexigrid, Smith & Nephew, Hull, UK), and a Velcro^®^ (Velcro, Manchester, NH, USA) loop ring surrounding the wound/wounds was sutured in place.

#### 4.3.3. Validation Study of Ischemic versus Nonischemic Wound Healing

This study included 16 of the rats, stratified by body weight into 2 groups: hydrogel (*n* = 8), and no hydrogel (*n* = 8). In 8 rats, both wounds received hydrogel (200 µL) covered by an occlusive, adhesive 2 cm × 4 cm dressing (Hydrosorb^®^ comfort, Hartmann, Heidenheim, Germany). The wounds in the remaining 8 rats were treated with the Hydrosorb^®^ comfort dressing alone. A protective nylon fabric lid was secured to the sutured Velcro^®^ loop. The experiment was concluded on postoperative day 6.

#### 4.3.4. Intervention Study in the Ischemic Wounds

This study included 28 rats, stratified by body weight into 3 groups: NPY (*n* = 9), IGF-1 (*n* = 9), and control hydrogel (*n* = 10).

Either 1 volume of NPY (5 mg/mL 10 mM HOAc) or 1 volume IGF-I (0.5 mg/mL 10 mM HOAc) was blended with 19 volumes of the hydrogel, and the control consisted of 1 volume of 10 mM HOAc blended with 19 volumes of the hydrogel. The final concentrations were 250 µg/mL for NPY and 25 µg/mL for IGF-I. The charged and control hydrogel blends were transferred to 4-mL polypropylene cryogenic vials (430491, Corning Life Sciences, Salt Lake City, UT, USA) labeled 1, 2, or 3, centrifuged (2000× *g*), and applied (200 µL) to the wounds by a positive displacement pipette with combitips (Multipipette^®^ plus, Eppendorf, Hamburg, Germany). The wounds were covered with 2 cm × 4 cm Hydrosorb^®^ comfort dressing. A protective nylon fabric lid was attached to the sutured Velcro^®^ loop. The experiment was concluded on postoperative day 14. Procedures are summarized in Table 2.

#### 4.3.5. Wound Care, Wound Area Measurements, and Calculation of Relative Wound Area Change

Every second day, rats were weighed, and wounds were cleansed with saline-moistened nonwoven swabs. The outer margins of the wounds were traced on transparent sheet. The wounds were redosed, and Hydrosorb^®^ comfort dressing was applied. From day 6 onwards (intervention study), gauze was applied over the Velcro^®^ device, and Tensoplast^®^ (BSN medical GmbH, Hamburg, Germany) was wrapped around the animals.

Wound areas were determined by the image analysis ImageJ software (1.46r, National Institutes of Health, Bethesda, MD, USA). The relative change (%) in wound area (R_i_) from day 0 was calculated as R_i_ = (A_i_ − A_Day 0_)/A_Day 0_ × 100, where A equals wound area and i equals postoperative day.

#### 4.3.6. Tissue Harvesting (Intervention Study)

Rats were euthanized by a blow to the head followed by cervical dislocation. Wounds with adjoining uninjured skin were excised, fixed in neutral 4% phosphate-buffered paraformaldehyde, and embedded in paraffin.

#### 4.3.7. In Situ Hybridization Assay of Y2R mRNA by RNAscope^®^ (Intervention Study)

Tissue sections were cut at 3 µm and deparaffinized. Proprietary reagents for pre-treatment, hybridization with the specific Rn-Npy2r probe (41448), and signal amplification were used according to the manufacturer (ACD, Bio-Techne, Abingdon, UK). The assay was validated in rat brain tissue [30], as shown in Figure 6.

#### 4.3.8. Histology (Intervention Study)

Tissue sections (3 µm) were cut onto microscopic glass slides, deparaffinized in xylene, rehydrated in a descending alcohol series, and stained with H&E, or with van Gieson. A senior consultant pathologist (L.M.R.G.) performed these investigations. Angiogenesis was estimated using a 4-point grading scale: 0 = absent; 1 = slight; 2 = moderate; 3 = pronounced. The overall degree of inflammation was assessed on a 4-point grading scale: 0 = absent; 1 = slight; 2 = moderate; 3 = pronounced. Additionally, extravascular lymphocytes, neutrophil granulocytes, plasma cells, and mast cells were counted in 3 different high-power (×400) fields of the granulation tissue; 2 at each wound margin and 1 in the middle of the wound. The various cell types were identified by morphological characteristics, described in, for example, standard textbooks [56]. The length and thickness of neoepithelium was measured at the longest and thickest part of the epithelial tongue. The presence of foreign bodies was graded on a 4-point scale: 0 = none; 1 = few; 2 = moderate; 3 = many. Multinucleated giant cells were counted. The slides were scanned at high resolution (NanoZoomer, C13220, Hamamatsu Photonics, Hamamatsu City, Japan) and the digitized images were processed by the NDP.View2 software (version 2.9.29, Hamamatsu Photonics, Hamamatsu City, Japan).

#### 4.3.9. Blinding

The blends were indistinguishable, and the identity of the hydrogels remained blinded to the caretakers of the animals, the operators, and the primary investigator until the experimental series was completed and results analyzed and interpreted. The histologic evaluations were performed without knowledge of group identity.

### 4.4. Statistical Analyses

Mean values were compared using one-way or two-way ANOVA followed by the Holm–Šídák test for multiple comparisons. Statistical tests were performed using Sigma-Plot 14.0. *p* < 0.05 was defined as statistically significant.

## 5. Conclusions

Ischemia delays wound healing, yet there are currently no treatments known to have documented beneficial effects. The neuropeptide NPY has been noted to promote angiogenesis, a necessary step for the improvement of ischemic wound healing. In this paper, we examined the impact of exogenous NPY in a preclinical model of ischemic wound healing in the skin. While NPY appeared to enhance the formation of new blood vessels in the granulation tissue, this enhancement did not result in higher rates of wound healing.

## Figures and Tables

**Figure 1 ijms-25-03346-f001:**
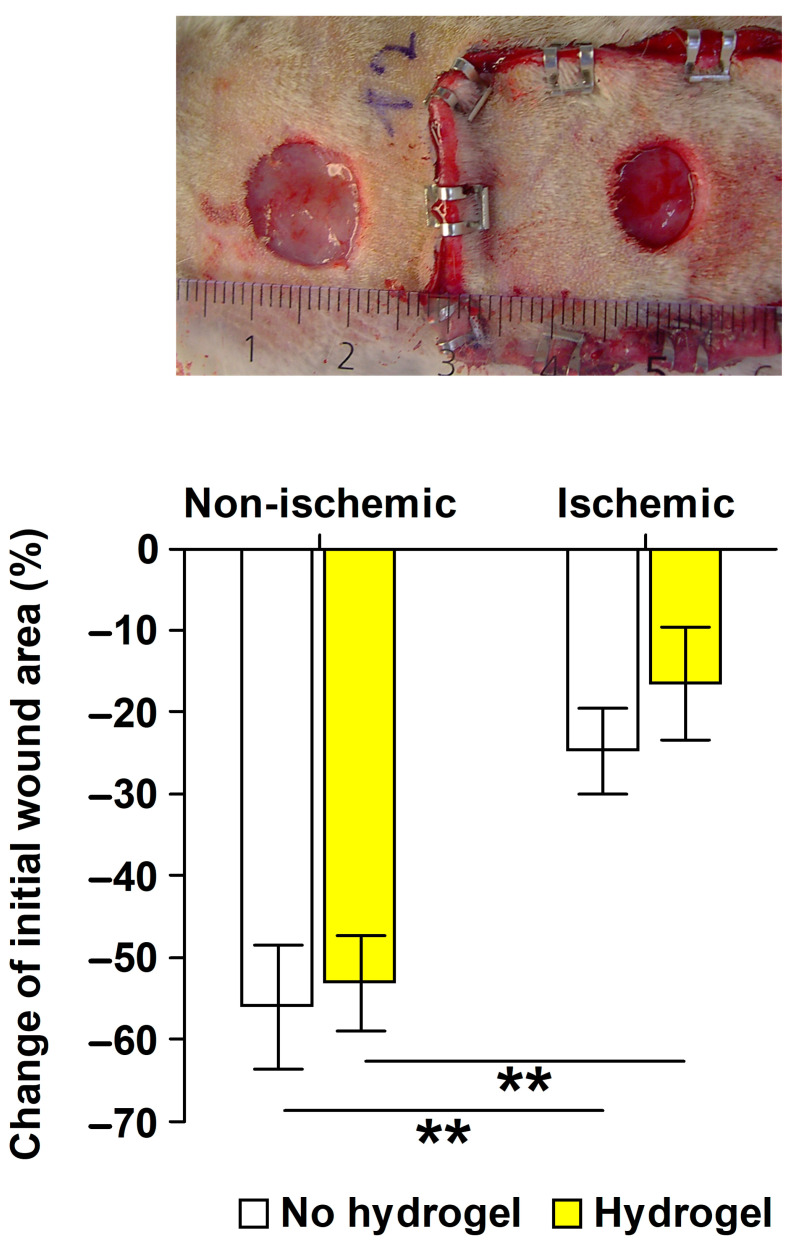
Validation of the ischemic wound model [21,26]. The nonischemic wound was made cephalad of the pedunculated ischemic skin flap (fixed with stainless steel clips) on the shaved back of a rat on day 0 (before application of hydrogel). Ruler (mm) is included in the image. The nonischemic wound was 104 mm^2^ and the ischemic wound 82 mm^2^ day 0. In the bar graph, wound area changes values on postoperative day 6 represent the eight rats treated without the hydrogel and the eight rats treated with the hydrogel. Negative values represent decreased wound size from day 0. Mean ± SEM (*n* = 8). One-way ANOVA followed by Holm–Šídák test for multiple comparisons; ** *p* < 0.01.

**Figure 2 ijms-25-03346-f002:**
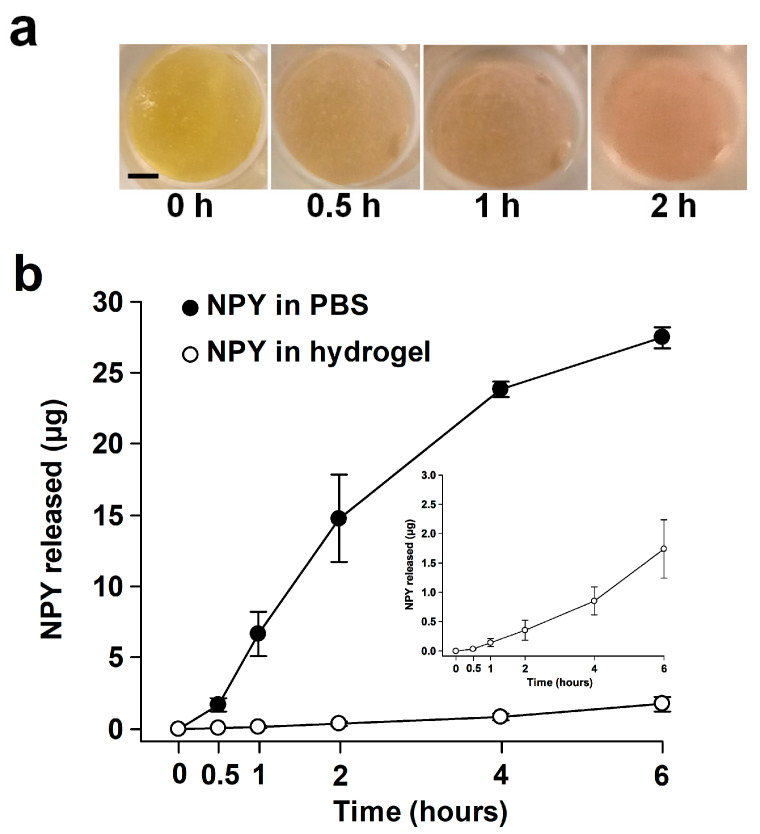
Release experiment with the Franz Cell. (**a**) NPY hydrogel formulation in donor chamber showing the shift of pH from acidic to neutral pH (phenol red indicator) with time of incubation. Scale bar, 2 mm. (**b**) Release of NPY in PBS (closed circles) compared with NPY in hydrogel (open circles) as a function of time. Inset: Close-up of the release of NPY in hydrogel. Mean ± SEM (*n* = 3).

**Figure 3 ijms-25-03346-f003:**
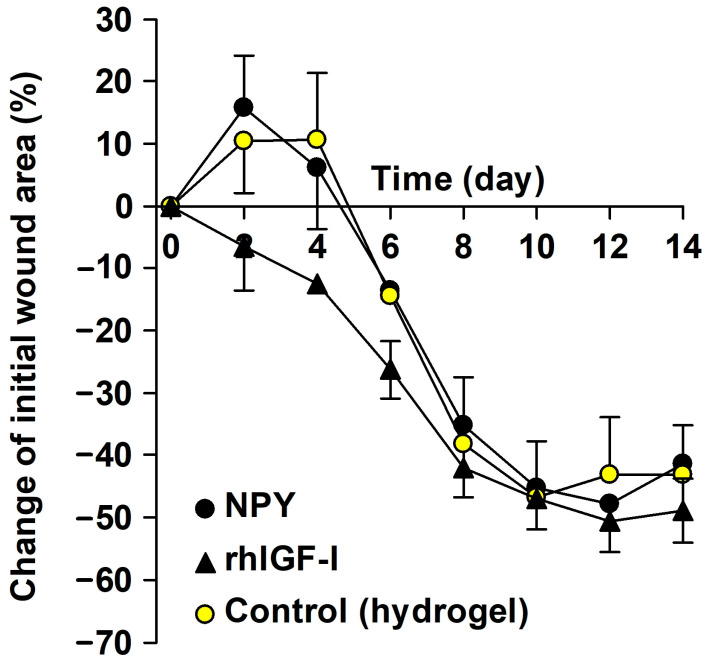
Relative change of wound area of the ischemic wounds in the NPY (*n* = 9), rhIGF-I (*n* = 9) and control (*n* = 10) groups. The initial wound area on day 0 was 49 ± 2.1 mm^2^ in the NPY group, 57 ± 3.5 mm^2^ in the rhIGF-I group, and 54 ± 3.7 mm^2^ in the control group. Negative values represent decreased wound size from day 0. Mean ± SEM.

**Figure 4 ijms-25-03346-f004:**
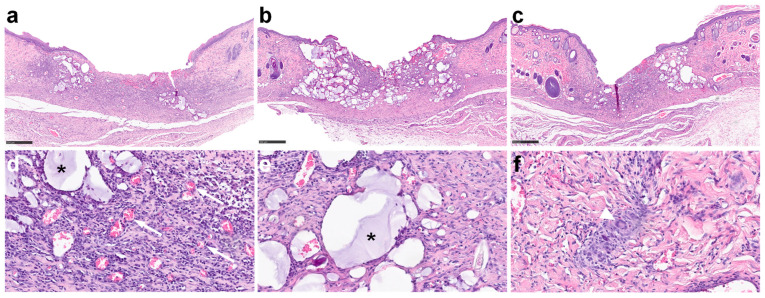
Microphotographs of full-thickness wounds on day 14 from the NPY (**a**,**d**), IGF-I (**b**,**e**), and control (**c**) groups at low magnification (**a**–**c**) and granulation tissue at higher magnification (**d**,**e**). Note neoangiogenesis of small vessels filled with erythrocytes (examples of blood vessels are indicated by white arrows) in the NPY group. Inflammatory cellular infiltrate consisting of mononuclear small lymphocytes and neutrophil granulocytes with segmented nuclei (**d**). Deposition of foreign material (*) is indicated (**d**,**e**). Arrowhead points to multinucleated giant cell (**f**). Scale bars, 500 µm (**a**–**c**). H&E stain.

**Figure 5 ijms-25-03346-f005:**
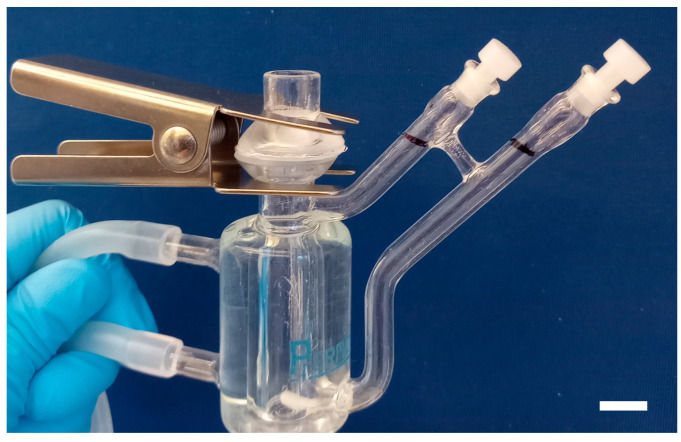
Franz Cell (PermeGear) used in the studies on NPY release. Scale bar, 10 mm.

**Figure 6 ijms-25-03346-f006:**
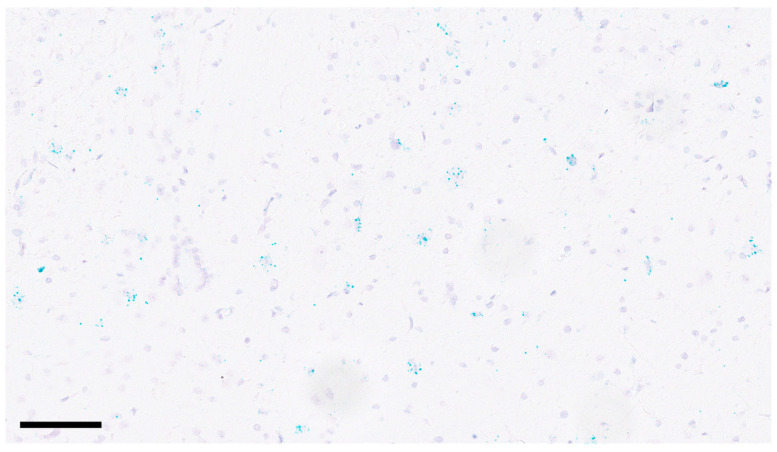
In situ hybridization reveals the presence of *Y2R* mRNA in several neurons of the brain of a male rat (280 g). Scale bar, 100 µm.

**Table 1 ijms-25-03346-t001:** Histological analysis of wounds after 14 days of treatment.

Group	Angiogenesis ^1^	Inflammatory Cell Infiltrate					Neoepithelium (mm) ^2^		Foreign Bodies ^4^	Giant Cells
		Grade ^1^	LC ^3^	NG ^3^	PC ^3^	MC ^3^	Ingrowth	Thickness		
NPY	3	3	316	85	6	3	1.1	0.14	1	3
IGF-I	1	1	144	42	1	0	1.4	0.061	3	1
Control	2	1.5	247	75	8	3	0.88	0.10	2	0

^1^ Assessed on a 4-point scale: 0 = absent; 1 = slight; 2 = moderate; 3 = pronounced. ^2^ Measured at the longest and thickest part of the epithelial tongue. ^3^ Number of cells in three high-power fields of view. ^4^ Assessed on a 4-point scale: 0 = none; 1 = few; 2 = moderate; 3 = many. LC, lymphocytes; MC, mast cells; NG, neutrophil granulocytes; PC, plasma cells.

**Table 2 ijms-25-03346-t002:** Summary of the procedures in the intervention study.

Procedure	Day							
	0	2	4	6	8	10	12	14
Wounding ^1^	✕							
Treatment ^2^	✕	✕	✕	✕	✕	✕	✕	
Wound tracing ^3^	✕	✕	✕	✕	✕	✕	✕	✕

^1^ One 8 mm full-thickness wound was made in the central pedunculated dorsal ischemic skin flap (3 cm × 7 cm). ^2^ NPY (50 µg/wound) and IGF-I (5 µg/wound) were delivered to the wounds in 200 µL of an amorphous hydrogel. Control wounds were treated with 200 µL of the amorphous hydrogel. ^3^ Wounds were traced (outer margins) 30 min after wounding day 0 and on subsequent days after cleansing the wound with saline-moistened nonwoven swab. ✕, action.

## Data Availability

The data presented in this study are available on request from the corresponding author.

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
