# Peer review of "Topical Neuropeptide Y for Ischemic Skin Wounds"

_ijms, 2024, doi:10.3390/ijms25063346_

Round 1

Reviewer 1 Report

Comments and Suggestions for Authors

The aim of the work was to explain the release of NPY in an amorphous hydrogel in vitro using a Franz diffusion cell. Furthermore, attempts have been made to determine the effects of NPY in vivo on ischemic wound healing in a validated preclinical model. Insulin-like growth factor-I (IGF-I), known to promote wound healing in this model, was used as a positive control.

It has been shown that the hydrogel was selected due to its capacity to prolong NPY release (p < 0.001), as demonstrated in a Franz diffusion cell. In the animals, an 8-mm full-thickness wound was made in a pedunculated dorsal ischemic skin flap. Wounds were treated and assessed for 14 days, and procured at the end of the experiment for in situ hybridization analysis (RNAscope®) targeting NPY receptor Y2R and for meticulous histologic examination. Wound healing rates, i.e. the percentage wound area changes, did not show an increase with NPY (p = 0.938) but did with rhIGF-I (p = 0.039) compared to the control. Y2R mRNA was not detected in the wounds or adjacent skin but was identified in the rat brain (positive control). Light microscopic examination revealed trends of increased angiogenesis and enhanced inflammatory cell infiltration with NPY compared to control. It was found the presence of melanophages in the wounds. The findings suggest the potential of NPY to enhance neovascularization under ischemic wound healing conditions, but further optimization of the carrier and dosage is necessary. The mechanism is elusive but likely involves NPY receptor subtypes other than Y2R.

The work provides very valuable information regarding attempts to use NPY in regenerative medicine.

The manuscript lacks a Conclusion, this subsection should be added. Figure 6 needs to be corrected, the quality of the drawing makes it too illegible.

Author Response

Thank you very much for taking your time to review our manuscript.

We have the following responses to your concerns:

  • We have added a Conclusion section.

"Ischemia delays wound healing, yet there are currently no treatments known to have documented beneficial effects. The neuropeptide NPY has been noted to promote angiogenesis, a necessary step for the improvement of ischemic wound healing. In this paper, we examined the impact of exogenous NPY in a preclinical model of ischemic wound healing in the skin. While NPY appeared to enhance the formation of new blood vessels in the granulation tissue, this enhancement did not result in higher rates of unusually long or complex."

  • The Figure 6 has been replaced by an image with higher resolution.

We hope these amendments are satisfactory.

Reviewer 2 Report

Comments and Suggestions for Authors

The manuscript is very well written and clearly presented.  

The NPY was perhaps not as effective at promoting ischemic wound healing than had been hoped but it did have an effect in inducing angiogenesis in the granulation tissue.  

Figure 4 could be improved for the non-expert reader by including high power images of the LC, NG, PC, MC, Foreign bodies and giant cells along with a description of how to identify them.  Not many people would know how to do this and even in the higher power images of d-f this is still to low to make them out.  This is just a suggestion for the education of readers and is not required for the science of the manuscript.

There are a few instances where not quite the right word was used.

L271 hold should be held

L344 “Wound Toilet” I am not sure what is meant here and suspect an issue with translation to English

L354  Tissue “Procurement” I don’t think is the right word – tissue harvesting would be better.

Author Response

We appreciate your constructive comments to our manuscript.

  • Figure 4. We have added markers of oreign bodies, blood vessels and multinucleated giant cells in the high power images (Figure 4d-f). Regarding identification of cell types these analyses were also performed by an experienced senior consultant pathologist in a blinded manner. We added a reference to a histology textbook that describes the typical characteristics of these cells.
  • Line 271: hold was changed to held.

  •  

    Line 344: “Wound Toilet” was changed to "Wound Care"

  • Line 354 "Tissue Procurement" was changed to "Tissue Harvesting" and in Abstract procured was changed to collected.